# GHOST: Generalizable One-Shot Federated Graph Learning with Proxy-Based Topology Knowledge Retention

**Jiaru Qian** [* 1]  **Guancheng Wan** [* 1]  **Wenke Huang** [* 1]  **Guibin Zhang** [* 2]  **Yuxin Wu** [3]  **Bo Du** [1]  **Mang Ye** [† 1]

## Abstract

Federated Graph Learning (FGL) proposes an effective approach to collaboratively training Graph Neural Networks (GNNs) while maintaining privacy. Nevertheless, communication efficiency becomes a critical bottleneck in environments with limited resources. In this context, one-shot FGL emerges as a promising solution by restricting communication to a single round. However, prevailing FGL methods face two key challenges in the one-shot setting: 1) They heavily rely on gradual personalized optimization over multiple rounds, undermining the capability of the global model to efficiently generalize across diverse graph structures. 2) They are prone to overfitting to local data distributions due to extreme structural bias, leading to catastrophic forgetting. To address these issues, we introduce **GHOST**, an innovative one-shot FGL framework. In GHOST, we establish a proxy model for each client to leverage diverse local knowledge and integrate it to train the global model. During training, we identify and consolidate parameters essential for capturing topological knowledge, thereby mitigating catastrophic forgetting. Extensive experiments on real-world tasks demonstrate the superiority and generalization capability of GHOST. The code is available at https: //github.com/JiaruQian/GHOST.

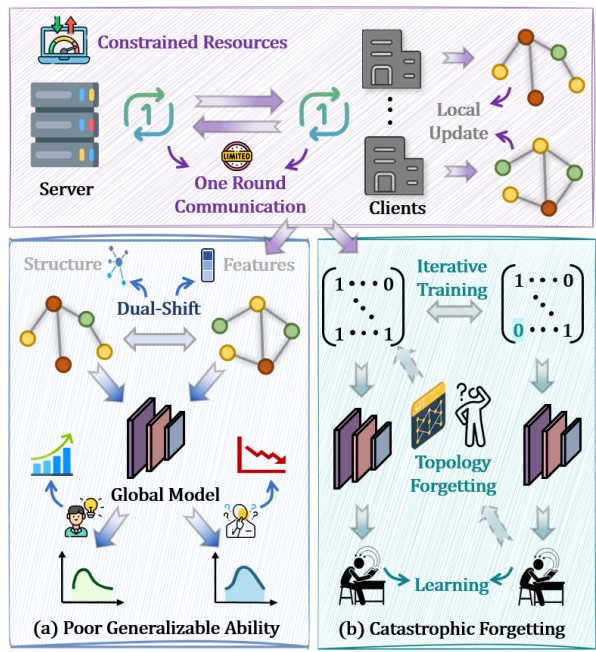

*Figure 1.* **Problem Illustration.** In environments with constrained resources, one-shot FGL becomes a promising solution by limiting communication to one round. We identify two key challenges in the one-shot setting. *(a) Poor Generalizable Ability:* Traditional FGL methods depend on gradual personalized updates. However, pattern shifts in both feature and structure dimension and limited communication rounds hinder the ability of the global model to efficiently generalize across diverse graphs. *(b) Catastrophic Forgetting:* Structural biases across clients cause the model to overfit to current local data distributions, leading to the loss of critical topology knowledge acquired previously.

## 1. Introduction

Federated Learning (FL) (Li et al., 2020a; Kairouz et al., 2021; Konečný, 2016; Hu et al., 2023) enables decentralized

---
[*]Equal contribution  [1]National Engineering Research Center for Multimedia Software, School of Computer Science, Wuhan University, Wuhan, China [2]National University of Singapore, Singapore [3]Renmin University of China. Correspondence to: Mang Ye <yemang@whu.edu.cn>.

*Proceedings of the 42nd International Conference on Machine Learning*, Vancouver, Canada. PMLR 267, 2025. Copyright 2025 by the author(s).

model training across multiple clients while ensuring data privacy by avoiding the aggregation of raw data. However, many real-world datasets are graph-structured (Xia et al., 2021; Wang et al., 2021; Li et al., 2025b; Wan et al., 2025b; 2024b), where nodes represent entities and edges capture the relationships or connections between them. In the context of graph-structured data, the integration of Graph Neural Networks (GNNs) (Wu et al., 2020; Scarselli et al., 2008; Li et al., 2025a; Wan et al., 2025a; Huang et al., 2023a) has given rise to Federated Graph Learning (FGL) . FGL has applications across various domains, including healthcare (Xu

et al., 2021; Antunes et al., 2022), social networks (Khan et al., 2021; He et al., 2019), and recommendation systems (Gao et al., 2023; Wu et al., 2021). The primary challenge in FGL lies in addressing the heterogeneity (Huang et al., 2023c; Sun et al., 2024; Liao et al., 2024), of graph data, especially in topology dimension. Recent studies (Huang et al., 2025; Tan et al., 2025) have proposed various strategies to overcome the challenge. Approaches such as topology-aware aggregation (Fu et al., 2025; Li et al., 2024d; Fu et al., 2024), graph simplification techniques (Chen et al., 2021; Lei et al., 2023), and knowledge distillation (Huang et al., 2023b; Zhu et al., 2024b) have demonstrated their potential to improve model alignment across clients.

Despite the effectiveness of these methods in traditional FGL scenarios, they face significant challenges when applied to environments with constrained communication resources, such as edge-computing systems or decentralized networks with limited bandwidth (Zhou et al., 2021). In such contexts, the efficiency of information exchange becomes a critical bottleneck, hindering the sharing and exploration of knowledge across clients during training. As communication costs escalate, the ability to transfer meaningful updates between clients is compromised, leading to performance degradation and ultimately impeding the learning process (Yao et al., 2024). A promising solution to this issue is one-shot FGL, which streamlines the communication process by reducing it to a single round. By minimizing the need for repeated communication, one-shot FGL can alleviate the strain on communication resources while still enabling effective model training, potentially overcoming the limitations of traditional multi-round methods.

However, existing FGL methods largely depend on personalized updates and multiple iterative communication rounds, focusing primarily on optimizing local models tailored to client data distributions (Chen et al., 2022; Zhang et al., 2023). While effective, personalized approaches are prone to overfit to local data, and the one-round communication setting further limits opportunities for iterative knowledge exchange. As a result, the chance to gradually capturing local diverse knowledge across multiple rounds is constrained, preventing models from efficiently refining generalizable knowledge. Consequently, the aggregated global model often struggles to generalize effectively to diverse graph data. This raises the following question: **I)** *How can we efficiently capture diverse local knowledge to train a generalized model in one communication round?*

In addition, under conditions of extreme structural bias, existing FGL models face a significant challenge in retaining previously learned structural knowledge while maintaining generalization capacity. Structural bias (Kong et al., 2024; Zhang et al., 2021), characterized by substantial differences in graph topologies across clients, can cause the model to overfit to the current data distributions encountered during training. As a result, critical topology knowledge acquired during previous training epochs are gradually forgotten, culminating in catastrophic forgetting (Serra et al., 2018; Lee et al., 2017; Kemker et al., 2018). This limitation not only reduces effectiveness of the model on previously encountered data but also impairs its ability to generalize across diverse graph structures. This leads to another crucial question: **II)** *how can we alleviate catastrophic forgetting while ensuring generalization across diverse graph structures?*

To address the aforementioned questions, we introduce **GHOST: G**eneralizable One-**SHO**t Federated Graph Learning with Proxy-Ba**S**ed **T**opology Knowledge Retention. To address **I)** , we first adopt a one-shot generalized FGL approach. For each client, we establish a proxy model to extract diverse knowledge from its local data. Specifically, we construct Dual-Level Alignment (DLA) to train the proxy model, enabling it to fully explore both feature and structural information. Each client uploads its proxy model parameters to the central server ***only once***. The acquired knowledge is then integrated to collaboratively train a GNN, enhancing its generalization capability while mitigating the negative impact of inherent data heterogeneity.

Since topological information plays a crucial role during training, we propose Topology-Conscious Knowledge Retention (TCKR) to address **II)**. Our approach explicitly explores partial structures within the input graph by computing the Topology-Consistency Criterion, which identifies the parameters essential for capturing topological information. By stabilizing these critical parameters from diverse graphs, we enable the model to consolidate acquired knowledge while balancing the integration of information under extreme structural bias, thereby alleviating catastrophic forgetting. Our principal contributions are summarized as follows.

- We are the first to incorporate the one-shot setting into generalized FGL. We identify that existing FGL methods face challenges in constrained communication scenarios, where they struggle to efficiently train a generalized global model. Additionally, catastrophic forgetting hampers the retention of topological knowledge across diverse graphs.
- We introduce GHOST, a generalized one-shot FGL approach. For each client, we introduce a proxy model to fully explore dual-level knowledge. Furthermore, we emphasize the retention of acquired topological information during global training to mitigate catastrophic forgetting.
- Extensive experiments validate the superiority and robust generalization capability of GHOST, demonstrating its adaptability across graphs of varying scales.

## 2. Related Work

**Federated Graph Learning.** Federated Graph Learning (FGL) enables collaborative training among multiple clients

via a trusted server without sharing graph data (He et al., 2021; Liu et al., 2024; Li et al., 2024c; Huang et al., 2023b; Wan et al., 2025c; Zhang et al., 2024a). FGL approaches are categorized into two types: inter-graph and intra-graph (Li et al., 2025c; Tan et al., 2025). Inter-graph methods such as GCFL+ (Xie et al., 2021) and FedStar (Tan et al., 2023) assume clients possess disjoint graphs, while intra-graph methods like FedSSP (Tan et al., 2024b) and FGGP (Wan et al., 2024a) handle clients that hold subgraphs of a global graph. However, these methods rely on substantial iterative communication, incurring significant costs. We are the first to address this issue by establishing a proxy model for each client to capture the knowledge from local data. Furthermore, each client uploads the parameters of its proxy model only once, enhancing both efficiency and security.

**One-shot Federated Learning.** One-shot Federated Learning (Guha et al., 2019; Zeng et al., 2024) has emerged as an efficient solution by minimizing communication rounds and enhancing security against eavesdropping attacks. Recent methods (Zhang et al., 2022; Heinbaugh et al., 2023; Jhunjhunwala et al., 2024; Yang et al., 2024) utilize knowledge distillation or neuron-matching techniques to optimize the global model. However, existing one-shot FL methods are primarily limited to computer vision tasks, overlooking graph structures, which results in significantly poorer performance and slower convergence in graph-based applications. In this study, we take an innovative approach by focusing on the multifaceted topological characteristics (Li et al., 2024b; Shi et al., 2024; Deng et al., 2025) of graphs, addressing feature-structure dual heterogeneity across clients.

**Catastrophic Forgetting.** Catastrophic forgetting is a critical issue in continual learning, where models continuously learn from a stream of data with the aim of gradually extending acquired knowledge for future learning (Kirkpatrick et al., 2017; Wang et al., 2024). Existing approaches (Wang et al., 2020b; Liu et al., 2021; Sun et al., 2023) to mitigate catastrophic forgetting can be broadly categorized into three paradigms: replay methods, regularization-based methods, and parameter isolation methods. A persistent challenge across these methodologies lies in balancing the integration of knowledge from varying data distributions (Huang et al., 2022), particularly for graph data in non-Euclidean spaces. To address this, our work stabilizes the key parameters involved in learning topological properties across diverse graphs, thereby alleviating catastrophic forgetting.

## 3. Motivation

### 3.1. Preliminaries

**Graph Neural Networks.** Define the graph data as $\mathcal{G} = (\mathcal{V}, \mathcal{E})$ with $\mathcal{V}$ as the node-set encompassing $N$ nodes and $\mathcal{E}$ as the edge-set. The feature matrix $\mathbf{X} = \{x_1, x_2, \ldots, x_N\}^\top$

compromises $F$-dimensional feature vectors $x_i$ corresponding to node $v_i$. The adjacency matrix of $\mathcal{G}$ is denoted as $\mathbf{A} \in \mathbb{R}^{N \times N}$, where $\mathbf{A}(i,j) = 1$ if there is an edge between nodes $i$ and $j$ and $\mathbf{A}(i,j) = 0$ otherwise. Then, Graph Neural Networks (GNN) obtains a representation of the node by recursively aggregating and transforming the representations of its neighbors. Specifically, the hidden representation $h_i$ of node $v_i$ at the $l$-th layer is computed as:

$$h_i^{l+1} = \text{UPD}(h_i^l, \text{AGG}(\{h_j^l : v_j \in \mathcal{N}(v_i)\})), \quad (1)$$

where $h_i^l$ is the representation of node $v_i$ at $l$-th layer, $\mathcal{N}(v)$ denotes the neighbor nodes set of node $v_i$, $\text{AGG}(\cdot)$ aggregates the neighbor representations and $\text{UPD}(\cdot, \cdot)$ updates the representation of node $v_i$ given its representation and the aggregated neighbor representations at the previous layer. Specifically, $h_i^0 = x_i$.

**One-shot Federated Graph Learning.** In the federated setting, there is one server and a set of clients $\mathcal{C}$, with $|\mathcal{C}| = K$ clients (indexed by $k$). Each client stores a local graph dataset $\mathcal{G}^k = (\mathcal{V}^k, \mathcal{E}^k)$, and the adjacency matrix for the $k$-th client's graph $\mathcal{G}^k$ is denoted as $\mathbf{A}^k$. Each node $v_i \in \mathcal{V}^k$ has a feature vector $x_i^k$ and a label $y_i^k$. Within the system, each client has a local differentiable model parameterized by $\boldsymbol{w}^k$, and uploads its parameters $\boldsymbol{w}^k$ to the server. In contrast to traditional Federated Graph Learning (FGL), one-shot FGL restricts the communication to a single round.

## 4. Methodology

### 4.1. Overview.

The proposed GHOST can be decomposed into two components: Dual-Level Aligned Proxy Model and Topology-Conscious Knowledge Retention, corresponding to Sec. 4.2 and Sec. 4.3, respectively. In Sec. 4.2, we introduce a proxy model for each client that captures both feature-based and topological knowledge. Each proxy model captures the diverse local knowledge and uploads its parameters to the server only once, thereby enhancing communication efficiency. In Sec. 4.3, we construct the Coherence Factor and Topology-Consistency Criterion to evaluate the contribution of each parameter in preserving structural knowledge. During the global training phase, we adopt the Knowledge Retention Loss to stabilize those critical parameters, thus alleviating overfitting and catastrophic forgetting. The framework illustration of our method is shown in Figure 2.

### 4.2. Dual-Level Aligned Proxy Model

**Motivation.** In one shot setting, existing FGL methods often fail to efficiently leverage the underlying diverse knowledge of all the clients, leading to the poor generalizable ability of the global model. As a consequence, the inherent data heterogeneity in both feature and structure dimension will

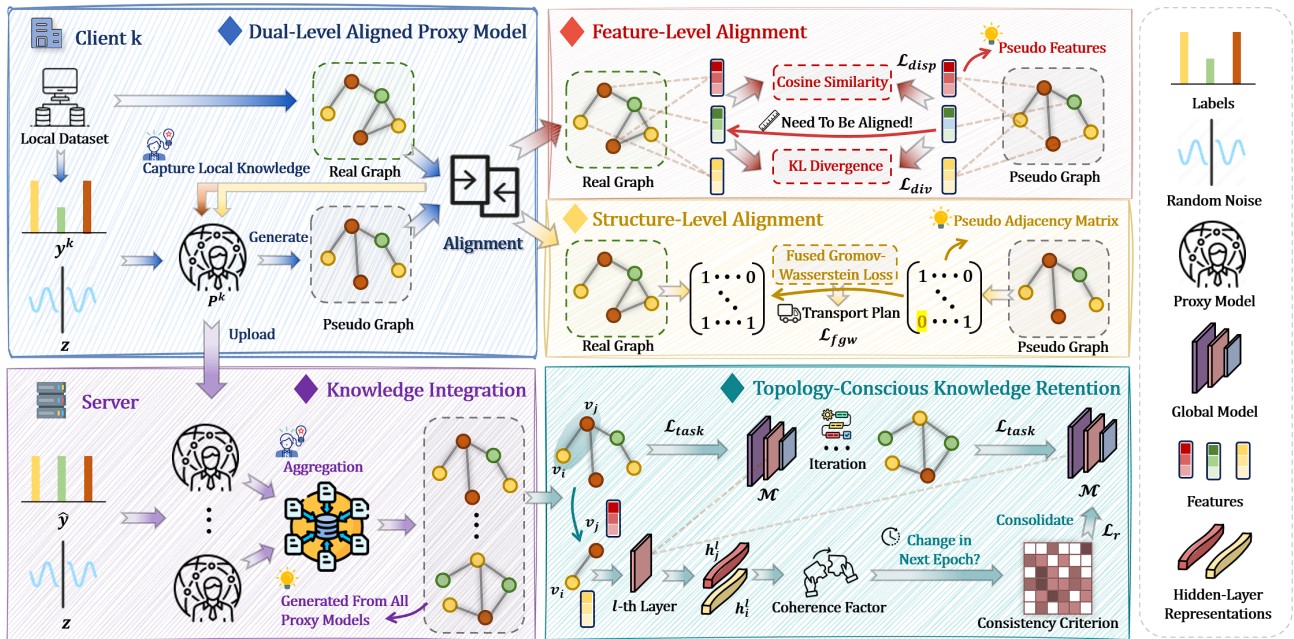

*Figure 2.* Architecture illustration of GHOST. The upper part represents client-side steps, while the lower part illustrates the server-side process. **Upper left** is the local training process where the proxy model of the client effectively captures local knowledge. **Upper right** shows the specific implementation of Dual-Level Alignment, which is divided into feature-level and structure-level. **Lower left** illustrates the server-side process of aggregating the knowledge learned by proxy models from all clients and generating an integrated knowledge set for training the global model. **Lower right** shows the global model training on downstream tasks with crucial parameters stabilized to alleviate catastrophic forgetting. The far-right section displays the legend. Zoom in for details.

inevitably undermine the model performance. To overcome this, we aim to develop a proxy model that efficiently captures diverse local knowledge and align the model in both feature and topological dual-level dimension with only one communication round.

**Local Proxy Model.** To fully capture the specific pattern of local graph data, we establish a proxy model $\boldsymbol{P}(\cdot)$ for each client. For each node $v_i \in \mathcal{V}$ in the graph $\mathcal{G}$, we input random noise $z$ sampled from a Gaussian distribution, along with the label $y_i$ of the node. The proxy model then generates a new feature vector $\hat{x}_i \in \mathbb{R}^F$:

$$\hat{x}_i = \boldsymbol{P}(z; y_i), \tag{2}$$

In the same way, we can obtain the pseudo feature matrix $\hat{\mathbf{X}} = [\hat{x}_1, \hat{x}_2, \ldots, \hat{x}_N]^\top \in \mathbb{R}^{N \times F}$. Based on $\hat{\mathbf{X}}$, we utilize the *K-Nearest Neighbors* strategy to construct the pseudo adjacency matrix $\hat{\mathbf{A}}$:

$$\mathbf{H} = \sigma\left(\hat{\mathbf{X}}\hat{\mathbf{X}}^\top\right), \hat{\mathbf{A}}(i,j) = \begin{cases} 1, & \text{if } j \in \text{TopK}\big(\mathbf{H}(i)\big), \\ 0, & \text{otherwise,} \end{cases} \tag{3}$$

where $\sigma(\cdot)$ represents the sigmoid function, $\text{TopK}(\cdot)$ selects the indices of the $\mathcal{K}$ largest values from a given input vector, and $\mathbf{H}(i)$ denotes the $i$-th row vector of $\mathbf{H}$. Subsequently, we obtain the pseudo graph $\hat{\mathcal{G}}$ with the pseudo feature matrix

$\hat{\mathbf{X}}$ and pseudo adjacency matrix $\hat{\mathbf{A}}$, while their corresponding labels $y = [y_1, y_2, \ldots, y_N]^\top$ remain unchanged.

**Dual-Level Alignment.** For graph data residing in non-Euclidean spaces, structure plays an indispensable role, encapsulating a wealth of unique information that is crucial for understanding the relationships and dynamics within the data. Thus, our objective is to ensure that the proxy model of each client thoroughly captures both feature-based and topological information from local data.

To facilitate the accurate representation of statistical properties in the proxy model, it is essential to align the probability distributions between the local data and the pseudo graph. To address this challenge, we introduce the Divergence Loss, a novel loss function designed to minimize the discrepancy between the probability distributions of the local data and the pseudo graph. This loss ensures that the proxy model maintains consistency with the underlying data distribution, thereby improving its ability to capture both feature-based and topological information.

$$\mathcal{L}_{div} = \frac{1}{|\mathcal{V}|} \sum_{i=1}^{|\mathcal{V}|} \sum_{f=1}^{F} s_i(f) \log \frac{s_i(f)}{\hat{s}_i(f)}, \tag{4}$$

where $s_i = \text{softmax}(x_i)$, $\hat{s}_i = \text{softmax}(\hat{x}_i)$ and $f$ denotes the $f$-th dimension of the vector. Moreover, to ensure that

the proxy model effectively aligns the local feature representation while mitigating the risk of mode collapse, we apply the Dispersion Loss to the proxy model:

$$\mathcal{L}_{disp} = 1 - \frac{1}{|\mathcal{V}|}\sum_{i=1}^{|\mathcal{V}|}\frac{\hat{x}_i \cdot x_i}{||\hat{x}_i||_2 \cdot ||x_i||_2}. \qquad (5)$$

Meanwhile, capturing the topological essence of graph data is equally important. To address this, we incorporate the Fused Gromov-Wasserstein (FGW) Loss, which combines both feature and structural alignment. This ensures that the proxy model not only aligns with the statistical properties of the local features but also preserves the intrinsic topological relationships encoded in the graph structure. Specifically, the FGW Loss is defined as:

$$\mathcal{L}_{fgw} = \min_{\mathbf{\Gamma}(\mu,\hat{\mu})}\sum_{i,j,u,v}\Big(a\big(\mathbf{A}(i,j) - \hat{\mathbf{A}}(u,v)\big)^2 \\ + (1-a)\|\mathbf{X}(i) - \hat{\mathbf{X}}(u)\|_2^2\Big)\mathbf{\Gamma}_{i,u}\mathbf{\Gamma}_{j,v}, \qquad (6)$$

where $a \in [0,1]$ and $\mathbf{X}(i)$ represents the $i$-th row vector of $\mathbf{X}$. The transport plan $\mathbf{\Gamma} \in \mathbb{R}^{N \times N}$ is optimized to align nodes between the local graph $\mathcal{G}$ and the pseudo graph $\hat{\mathcal{G}}$, such that $\mathbf{\Gamma}_{i,u}$ quantifies the amount of mass transported between node $i$ in $\mathcal{G}$ and node $u$ in $\hat{\mathcal{G}}$. The optimization is constrained by the distributions $\mu$ and $\hat{\mu}$, which represent the marginal distributions of nodes in $\mathcal{G}$ and $\hat{\mathcal{G}}$, respectively. These are typically assumed to be uniform distributions, reflecting the equal importance of all nodes: $\sum_{j=1}^{N}\mathbf{\Gamma}_{i,j} = \mu(i)$, $\sum_{i=1}^{N}\mathbf{\Gamma}_{i,j} = \hat{\mu}(j)$. Thus, we establish the Dual-Level Alignment Loss as follows:

$$\mathcal{L}_{dual} = \mathcal{L}_{disp} + \lambda_d\mathcal{L}_{div} + \lambda_f\mathcal{L}_{fgw}, \qquad (7)$$

where $\lambda_d$ and $\lambda_f$ are hyperparameters that balance the contributions of each loss function. After local training with the Dual-Level Alignment Loss, each client uploads the parameters of its proxy model, restricting communication to a single round. On the server side, the diverse information gathered from all proxy models is aggregated to collaboratively train a generalizable global GNN, enabling it to effectively handle downstream tasks.

### 4.3. Topology-Conscious Knowledge Retention.

**Motivation.** Extreme structural bias in FGL models causes them to overfit the current data distribution, leading to catastrophic forgetting and poor generalization ability. To address this issue, we propose Topology-Conscious Knowledge Retention, which explicitly leverages the topological information in graphs. This approach helps the model consolidate acquired knowledge while balancing the integration of diverse information from heterogeneous graph structures.

**Knowledge Integration.** After local training, the server aggregates the proxy models to integrate diverse knowledge. Specifically, for the proxy model $\mathbf{P}^k(\cdot)$ from client $k$, we input random noise $\mathbf{z}$ sampled from a Gaussian distribution. To simulate real-world data diversity, we generate a pseudo-label vector $\hat{y}^k \in \mathbb{R}^N$, where the distribution across all classes is uniform, ensuring that all aspects of knowledge are considered. Then, by applying Equation (2) and Equation (3), we generate the pseudo graph $\hat{\mathcal{G}}^k$ with feature matrix $\hat{\mathbf{X}}^k$ and adjacency matrix $\hat{\mathbf{A}}^k$ through the proxy model $\mathbf{P}^k(\cdot)$. The knowledge set for client $k$ can be denoted as $\mathcal{G}_{\mathcal{S}}^k = \{\hat{\mathcal{G}}_1^k, \hat{\mathcal{G}}_2^k, \ldots, \hat{\mathcal{G}}_M^k\}$, where $M$ is the number of generated graphs. Subsequently, the integrated knowledge set is given by $\mathcal{G}_{\mathcal{S}} = \mathcal{G}_{\mathcal{S}}^1 \cup \mathcal{G}_{\mathcal{S}}^2 \cup \ldots \cup \mathcal{G}_{\mathcal{S}}^K$, which is used to jointly train the global model.

**Topology-Consistency Criterion.** Given the extensive knowledge generated by all proxy models, the global model faces the risk of overfitting to the current knowledge set and catastrophic forgetting. Therefore, it is crucial to extract topological commonalities across diverse graphs and consolidate the acquired information during the global training phase. Specifically, we aim to identify critical parameters for capturing topological consensus among the diverse knowledge in the integrated set.

For pseudo graph $\hat{\mathcal{G}}_m^k$ in knowledge set $\mathcal{G}_{\mathcal{S}}^k$, we denote the hidden representation of node $\hat{v}_i$ at the $l$-th layer as $\hat{h}_i^l$. Then, we introduce Coherence Factor $\gamma_{ij}$ to measure the alignment between the hidden representations of connected nodes $v_i$ and $v_j$, which is computed as:

$$\gamma_{ij}^l = (\hat{h}_i^l \cdot \boldsymbol{\theta}_{\boldsymbol{w}}^l)^\top \cdot \tanh(\hat{h}_j^l \cdot \boldsymbol{\theta}_{\boldsymbol{w}}^l), \qquad (8)$$

where $\boldsymbol{\theta}$ represents the parameters of the global GNN, and $\boldsymbol{\theta}_{\boldsymbol{w}}^l$ denotes the weight matrix of the $l$-th layer. Then, we obtain the vector $\gamma_i^l = [\gamma_{i1}^l, \gamma_{i2}^l, \ldots, \gamma_{in}^l]$ for node $v_i$, where $n = |\mathcal{N}(v_i)|$. The Coherence Factor $\gamma_{ij}^l$ quantifies the similarity between the acquired representations of two connected nodes, thereby encouraging the model to maintain the memory of consistent topological structures across different graphs in the integrated knowledge set.

As demonstrated in (Zenke et al., 2017), the infinitesimal update $\delta(\boldsymbol{\theta})$ in the parameters $\boldsymbol{\theta}$ during the training phase leads to a corresponding change $\delta(\gamma_{ij})$ in the Coherence Factor. Building on this relationship, we compute the Topology-Consistency Criterion to assess the importance of each parameter in capturing the structural knowledge of the input graphs. The Topology-Consistency Criterion $\mathcal{T}_l^k$ for each knowledge set $\mathcal{G}_{\mathcal{S}}^k$ can be calculated as:

$$\mathcal{T}_l^k = \left[\left\|\frac{\partial}{\partial\boldsymbol{\theta}_{\boldsymbol{w_i}}^l}\left(\sum_{j=1}^{|\mathcal{V}|}\|\gamma_j^l\|_2^2\right)\right\|\right]_{i=1}^W, \qquad (9)$$

where $W = |\boldsymbol{\theta}_{\boldsymbol{w}}^l|$ denotes the number of weight parame-

ters. The Criterion $\mathcal{T}_l^k$ has the same shape as the weight matrix $\boldsymbol{\theta}_{\boldsymbol{w}}^l$, representing the importance of each parameter in capturing the structural knowledge. We then define $\mathcal{T}^k = [\mathcal{T}_1^k, \mathcal{T}_2^k, \ldots, \mathcal{T}_{L-1}^k]$ as the criterion for the entire model with $L$ layers (excluding the projection head).

**Knowledge Retention.** When the global model transitions to a new training phase with the knowledge set $\mathcal{G}_\mathcal{S}^{k+1}$ after learning from $\mathcal{G}_\mathcal{S}^k$, it is crucial to stabilize the key parameters identified by the previous Topology-Consistency Criteria $\{\mathcal{T}^1, \mathcal{T}^2, \ldots, \mathcal{T}^k\}$. To achieve this, we introduce the Knowledge Retention Loss, which aims to consolidate the acquired topological knowledge:

$$\mathcal{L}_r^{k+1} = \sum_{i=1}^k \sum_{l=1}^{L-1} \mathcal{T}_l^k \otimes (\boldsymbol{\theta^l} - \boldsymbol{\theta}_i^{l*}), \qquad (10)$$

where $\otimes$ denotes element-wise multiplication, and $\boldsymbol{\theta}_i^{l*}$ represents the optimal parameters of the $l$-th layer after training on the knowledge set $\mathcal{G}_\mathcal{S}^i$. However, some values in $\mathcal{T}$ may become excessively large, reducing the flexibility of the model and its ability to assimilate new knowledge. To address this, we introduce the $l_2$ norm as a regularization term for $\mathcal{T}$, thereby enhancing the robustness and plasticity of the global model. Thus, we define the loss function for training the global GNN with the knowledge set $\mathcal{G}_\mathcal{S}^{k+1}$ as:

$$\mathcal{L}_{global}^{k+1} = \mathcal{L}_{task}^{k+1} + \lambda_r \mathcal{L}_r^{k+1} + \lambda_n ||\mathcal{T}^{k+1}||_2, \qquad (11)$$

where $\mathcal{L}_{task}$ represents the task-specific loss function. For instance, we can employ Cross-Entropy (CE) Loss or Negative Log-Likelihood (NLL) Loss. The hyperparameters $\lambda_r$ and $\lambda_n$ control the relative importance of each loss term.

### 4.4. Discussions on Privacy.

Privacy security plays a crucial role in FGL systems. In GHOST, we propose the Center-Shifting method to enhance privacy security. Specifically, the proxy model generates pseudo graphs using labels and random noise sampled from a Gaussian distribution. For simplicity, a standard normal distribution is typically used, where the center $\chi$ is set to 0. However, we introduce a shift $\epsilon$ to the center of the distribution, $\chi$, at the client side and communicate this shift $\epsilon$ to the server either offline or through encryption methods (Heinbaugh et al., 2023; Zhou et al., 2020). In our approach, each client only uploads the parameters of its proxy model to the server online once. We consider the worst-case scenario where an eavesdropping attacker intercepts all the parameters. However, without knowledge of the specific shift $\epsilon$, the attacker can only randomly select a center $\hat{\chi}$, which is likely to differ from the true center ($\chi + \epsilon$). Alternatively, the attacker might attempt to overlap the center using a wide uniform distribution, trying to align with the original distribution. Nevertheless, experiments conducted in Appendix G

prove that the attacker has no effective means to recover meaningful data through such guessing attempts.

## 5. Experiment

In this section, we comprehensively evaluate our proposed GHOST by addressing the following key questions.

- **Q1: Superiority.** Does GHOST maintain or surpass baseline performance?
- **Q2: Resilience** How does GHOST perform under varying degrees of data heterogeneity?
- **Q3: Effectiveness.** Do different modules of GHOST contribute to its overall performance?
- **Q4: Sensitivity.** How does GHOST perform under different hyper-parameter settings?

The answer of **Q1–Q3** are illustrated in Sec. 5.2-Sec. 5.4, and the analyses of **Q4** can be found in Appendix F.

### 5.1. Experimental Setup

We perform experiments on node classification tasks in various scenarios to validate the superiority of our framework.

**Datasets.** To effectively evaluate the performance of our approach, we employed seven benchmark graph datasets of various scales and features, including Cora (McCallum et al., 2000), CiteSeer (Giles et al., 1998), PubMed (Canese & Weis, 2013), Chameleon (Pei et al., 2020), Amazon-Photo , Coauthor-CS (Shchur et al., 2018) and Ogbn-Arxiv (Hu et al., 2020). Detailed descriptions and splits for these datasets can be found in Appendix B.

**Baselines.** We compare our method with several traditional FL Approaches: (1) **FedAvg** [ASTAT17] (McMahan et al., 2017), (2) **FedProx** [MLSys20] (Li et al., 2020b), (3) **FedNova** [NeurIPS20] (Wang et al., 2020a), (4) **FedRCL** [CVPR24] (Seo et al., 2024); three popular FGL approaches: (5) **FedPub** [ICML23] (Baek et al., 2023), (6) **FedTAD** [IJCAI24] (Zhu et al., 2024b), (7) **FedGTA** [VLDB24] (Li et al., 2024d); three One-shot FL methods: (8) **DENSE** [NeurIPS22] (Zhang et al., 2022), (9) **FedCVAE** [ICLR23] (Heinbaugh et al., 2023), (10) **FedSD2C** [NeurIPS24] (Zhang et al., 2024b). Detailed descriptions of all the baselines can be found in Appendix C.

**Implement Details.** Following the mainstream research practices, we utilize node classification as the downstream task. We set $|\mathcal{C}| = 10$ clients and draw $p_k \sim \text{Dir}(\alpha)$ from a Dirichlet distribution and assign a fraction $p_k^c$ of class $c$ to client $k$. Ablation Study on varying numbers of clients can be found in Appendix E. The parameter $\alpha$ governs the degree of non-IIDness, and we set $\alpha = 0.05$ to simulate the high heterogeneity scenario. For the alignment phase of each proxy model, we set the local training epoch

*Table 1.* **Comparison with the state-of-the-art methods** on seven real-world datasets. We report node classification accuracies (%) for downstream task performance. Green arrows ↑ denote advancements in accuracy metrics than FedAvg while red arrows ↓ indicate regressions. OOM means out-of-memory error. The best and second results are highlighted with **bold** and underline, respectively.

| Methods | Cora | CiteSeer | PubMed | Chameleon | Amz-Photo | Coauthor-CS | Obgn-Arxiv |
|---|---|---|---|---|---|---|---|
| FedAvg [ASTAT17] | 30.61 | 32.88 | 57.91 | 19.89 | 23.12 | 22.50 | 14.58 |
| **Traditional FL** | | | | | | | |
| FedProx [MLSys20] | 30.98$_{\uparrow 0.37}$ | 35.73$_{\uparrow 2.85}$ | 50.56$_{\downarrow 7.35}$ | 19.78$_{\downarrow 0.11}$ | 24.16$_{\uparrow 1.04}$ | 21.44$_{\downarrow 1.06}$ | 13.99$_{\downarrow 0.59}$ |
| FedNova [NeurIPS20] | 14.21$_{\downarrow 16.40}$ | 18.58$_{\downarrow 17.30}$ | 33.48$_{\downarrow 24.43}$ | 21.30$_{\uparrow 1.41}$ | 6.15$_{\downarrow 16.97}$ | 18.83$_{\downarrow 3.67}$ | 1.17$_{\downarrow 13.41}$ |
| FedRCL [CVPR24] | 17.60$_{\downarrow 13.01}$ | 12.73$_{\downarrow 20.15}$ | 28.12$_{\downarrow 29.79}$ | 18.48$_{\downarrow 1.41}$ | 4.92$_{\downarrow 18.20}$ | 14.75$_{\downarrow 7.75}$ | 2.56$_{\downarrow 12.02}$ |
| **Traditional FGL** | | | | | | | |
| FedPub [ICML23] | 30.52$_{\downarrow 0.09}$ | 34.91$_{\uparrow 2.03}$ | 41.22$_{\downarrow 16.69}$ | 17.61$_{\downarrow 2.28}$ | 21.91$_{\downarrow 1.21}$ | 26.75$_{\uparrow 4.25}$ | 10.02$_{\downarrow 4.56}$ |
| FedTAD [IJCAI24] | 30.43$_{\downarrow 0.18}$ | 33.86$_{\uparrow 0.98}$ | 39.32$_{\downarrow 18.59}$ | 20.11$_{\uparrow 0.22}$ | 22.01$_{\downarrow 1.11}$ | 14.09$_{\downarrow 8.41}$ | OOM$_{—}$ |
| FedGTA [VLDB24] | 14.02$_{\downarrow 16.59}$ | 17.75$_{\downarrow 15.13}$ | 31.45$_{\downarrow 26.46}$ | 21.20$_{\uparrow 1.31}$ | 4.10$_{\downarrow 19.02}$ | 10.80$_{\downarrow 11.70}$ | 1.15$_{\downarrow 13.43}$ |
| **One-shot FL** | | | | | | | |
| DENSE [NeurIPS22] | 12.92$_{\downarrow 17.71}$ | 7.87$_{\downarrow 25.01}$ | 20.84$_{\downarrow 37.07}$ | 19.90$_{\uparrow 0.01}$ | 4.93$_{\downarrow 18.19}$ | 3.96$_{\downarrow 18.54}$ | 0.33$_{\downarrow 14.25}$ |
| FedCVAE [ICLR23] | 30.89$_{\uparrow 0.28}$ | 34.76$_{\uparrow 1.88}$ | 52.01$_{\downarrow 5.90}$ | 21.74$_{\uparrow 1.85}$ | 31.62$_{\uparrow 8.50}$ | 14.60$_{\downarrow 7.90}$ | 13.71$_{\downarrow 0.87}$ |
| FedSD2C [NeruIPS24] | 17.78$_{\downarrow 12.83}$ | 29.96$_{\downarrow 2.92}$ | 26.12$_{\downarrow 31.79}$ | 22.72$_{\uparrow 2.83}$ | 8.73$_{\downarrow 14.39}$ | 3.88$_{\downarrow 18.62}$ | 0.76$_{\downarrow 13.82}$ |
| **Ours** | **50.41**$_{\uparrow 19.80}$ | **37.75**$_{\uparrow 5.87}$ | **58.87**$_{\uparrow 0.96}$ | **25.22**$_{\uparrow 5.33}$ | **37.22**$_{\uparrow 14.10}$ | **29.91**$_{\uparrow 7.41}$ | **17.24**$_{\uparrow 2.66}$ |

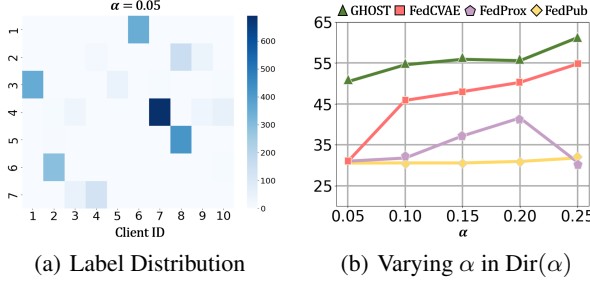

(a) Label Distribution     (b) Varying $\alpha$ in $\mathrm{Dir}(\alpha)$

*Figure 3.* **Resilience Study** of GHOST on the Cora dataset. In (a), we illustrate the label distribution under high data heterogeneity. In (b), we vary $\alpha$ within the range $[0.05, 0.25]$ with a step size of $0.05$ in $\mathrm{Dir}(\alpha)$, representing different degrees of data heterogeneity. Further details are provided in Sec. 5.3.

$T_L$ to 100. As for hyperparameters, $\lambda_d$ and $\lambda_f$ are determined through a grid search within $\{0.01, 0.05, 0.1, 0.5\}$ and $\{0.1, 0.2, 0.5, 1\}$ respectively. More implement details and parameter settings can be found in Appendix D.

## 5.2. Superiority

To address **Q1**, we analyze the superior performance of GHOST. We conduct the generalization setting where one global model is evaluated on test data of all clients. We demonstrate the node classification performance with different graph datasets and summarize the final test accuracy in Tab. 1. Notably, three key observations emerge: ❶ GHOST outperforms all other baselines in all datasets by fully capturing shared patterns among clients and consolidating the acquired knowledge. The dual-level knowledge obtained from all proxy models ensures the global model perform well across diverse data distributions. ❷ Algorithms tailored

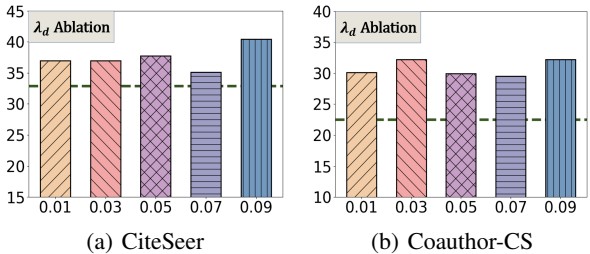

(a) CiteSeer      (b) Coauthor-CS

*Figure 4.* **Hyperparameter Analysis** on $\lambda_d$ on datasets CiteSeer and Coauthor-CS. The green dash denotes the baseline performance. See details and more ablation results in Appendix F.

for FGL such as FedPub and FedTAD perform better in most datasets than traditional FL methods due to their specific designs for graph-structured data in non-Euclidean space. However, they heavily rely on gradual personalized iterative updates that optimize models tailored to local data distributions. This leads to severe performance degradation in the generalized one-shot setting where they fail to efficiently capture all the local knowledge. ❸ One-shot FL approaches such as FedCVAE perform well in some small-scale graphs. However, their generative models ignore unique structural properties within the graph data, leading to poor performance in large-scale graphs with complex structures.

## 5.3. Resilience

To address **Q2**, we evaluate the performance of GHOST in comparison with other baselines under various degrees of data heterogeneity. Specifically, we manipulate $\alpha$ in the Dirichlet distribution $\mathrm{Dir}(\alpha)$ with values in $[0.05, 0.25]$ and a step size as $0.05$. A lower $\alpha$ corresponds to a more skewed label distribution across clients. To better illustrate it, we

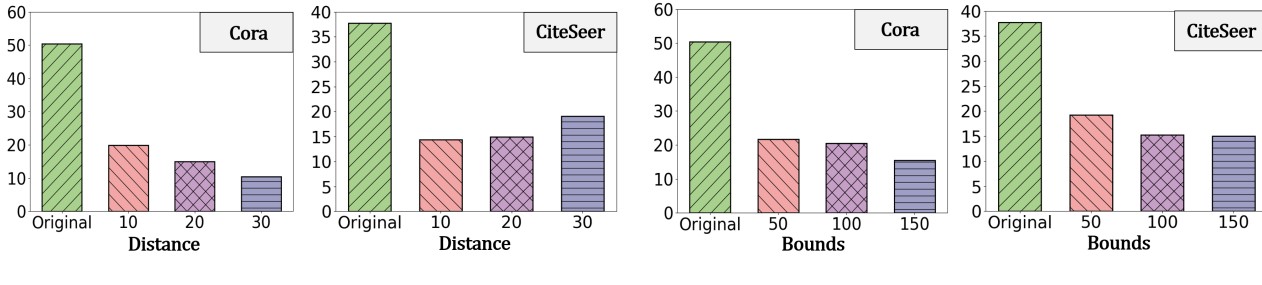

(a) Center Guess Attempt

(b) Overlapping Attempt

*Figure 5.* **Security Analysis** on Cora and CiteSeer. (a) shows the performance of the global model at varying distances from the shifted distribution center $(\chi + \epsilon)$. (b) presents the corresponding performance under different guesses based on the range of the uniform distribution. **Original** refers to the performance of the global model trained by the server, where the specific $\epsilon$ is communicated offline or via encryption methods securely. See Appendix G for more details.

*Table 2.* **Ablation study** of key modules (DLA, TCKR) in GHOST on Cora, CiteSeer, Chameleon and Coathor-CS. Green arrows ↑ denote advancements in accuracy metrics. The best results are highlighted with **bold**.

| DLA | TCKR | Dataset | | | |
| --- | --- | --- | --- | --- | --- |
| | | Cora | CiteSeer | Chameleon | Coauthor-CS |
| ✗ | ✗ | 12.65 | 14.46 | 17.17 | 4.32 |
| ✗ | ✓ | $15.12_{\uparrow 2.57}$ | $16.18_{\uparrow 1.72}$ | $21.52_{\uparrow 4.35}$ | $7.86_{\uparrow 3.54}$ |
| ✓ | ✗ | $46.93_{\uparrow 34.28}$ | $36.40_{\uparrow 21.94}$ | $25.00_{\uparrow 7.83}$ | $27.80_{\uparrow 23.48}$ |
| ✓ | ✓ | $\mathbf{50.41}_{\uparrow 37.76}$ | $\mathbf{37.75}_{\uparrow 23.29}$ | $\mathbf{25.22}_{\uparrow 8.05}$ | $\mathbf{29.91}_{\uparrow 25.59}$ |

*Table 3.* **Ablation study** of components $\{\mathcal{L}_{disp}, \mathcal{L}_{div}, \mathcal{L}_{fgw}\}$ in DLA module on Cora, CiteSeer, Chameleon and Coathor-CS. Green arrows ↑ denote advancements in accuracy metrics. The best results are highlighted with **bold**.

| $\mathcal{L}_{disp}$ | $\mathcal{L}_{div}$ | $\mathcal{L}_{fgw}$ | Dataset | | | |
| --- | --- | --- | --- | --- | --- | --- |
| | | | Cora | CiteSeer | Chameleon | Coauthor-CS |
| ✗ | ✗ | ✗ | 12.65 | 14.46 | 17.17 | 4.32 |
| ✓ | ✗ | ✗ | $27.31_{\uparrow 14.66}$ | $16.63_{\uparrow 2.17}$ | $20.11_{\uparrow 2.94}$ | $12.19_{\uparrow 7.87}$ |
| ✓ | ✓ | ✗ | $25.85_{\uparrow 13.20}$ | $28.31_{\uparrow 13.85}$ | $21.41_{\uparrow 4.24}$ | $13.85_{\uparrow 9.53}$ |
| ✓ | ✓ | ✓ | $\mathbf{46.93}_{\uparrow 34.28}$ | $\mathbf{36.40}_{\uparrow 21.94}$ | $\mathbf{25.00}_{\uparrow 7.83}$ | $\mathbf{27.80}_{\uparrow 23.48}$ |

use the Cora dataset as an example and visualize the label distribution across clients for $\alpha = 0.05$ through a heatmap in Figure 3(a). We compare GHOST with three strong baselines (FedProx [FL] , FedPub [FGL] , and FedCVAE [One-shot FL]) under different values of $\alpha$. The experimental results shown in Figure 3(b) indicate that GHOST consistently outperforms the baselines in all settings, demonstrating its strong resilience to data heterogeneity.

### 5.4. Effectiveness

To address **Q3**, we conduct an ablation study to evaluate the contributions of different modules in GHOST respectively. In GHOST, we first establish a proxy model for each client and perform Dual-Level Alignment (DLA) to fully explore local knowledge. During the global traing phase, we apply Topology-Conscious Knowledge Retention (TCKR) to consolidate previously acquired topological information. In this section, we utilize Cora, Citeseer, Amazon-Photo, and Coauthor-CS as example datasets, varying these components to assess their effectiveness.

**Effects of Modules in GHOST.** First, we perform an ablation study for DLA and TCKR in our framework. We adopt node classification as the downstream task. The experimental results are shown in Tab. 2. We observe that both DLA and TCKR significantly enhance model perfor-

mance. Notably, the introduction of the DLA module leads to substantial accuracy improvements across all settings. This demonstrates that the DLA module effectively leverages the embedded knowledge in data and aligns the proxy model's understanding with local data at both feature and structural levels. Motivated by this, we further explore the key components of the DLA module.

**Effects of Components in DLA.** To further investigate the role of components within DLA, we conduct an ablation study by varying the three alignment loss functions $\{\mathcal{L}_{disp}, \mathcal{L}_{div}, \mathcal{L}_{fgw}\}$, excluding the TCKR module to isolate the effects of these components. The experimental results are shown in Tab. 3. We observe that all three components contribute to the alignment process. $\mathcal{L}_{disp}$ and $\mathcal{L}_{div}$ focus on feature-level consistency, while $\mathcal{L}_{fgw}$ provides significant improvements by emphasizing structural alignment. This underscores the importance of structure in Federated Graph Learning systems.

## 6. Conclusion

In this paper, we are pioneers in addressing two key challenges of existing FGL approaches in the one-shot setting: their poor generalizable capability and catastrophic forgetting. We introduce **GHOST**: **G**eneralizable One-S**HO**t Federated Graph Learning with Proxy-Ba**S**ed **T**opology

Knowledge Retention. First, we propose a proxy model for each client, enabling the capture local knowledge in both feature and structural dimension. Then, the integration of knowledge from all clients into a global model enhances its ability to generalize across diverse graph structures. To mitigate catastrophic forgetting, we compute the Topology-Consistency Criterion to identify and consolidate parameters crucial for extracting underlying topological knowledge. Extensive experiments on real-world datasets demonstrate the effectiveness and generalization capability of GHOST in high data heterogeneity scenarios, offering a promising direction for scalable FGL applications in practical setting.

## Acknowledgement.

This research is supported by the National Key Research and Development Project of China (2024YFC3308400), the National Natural Science Foundation of China (Grants 62361166629, 62176188, 623B2080), the Wuhan University Undergraduate Innovation Research Fund Project. The supercomputing system at the Supercomputing Center of Wuhan University supported the numerical calculations in this paper.

## Impact Statement

This paper presents work whose goal is to advance the field of Machine Learning. There are many potential societal consequences of our work, none of which we feel must be specifically highlighted here.

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

# A. Notations.

We present a comprehensive review of the commonly used notations and their definitions in Tab. 4.

*Table 4.* Notation and Definitions

| Notation | Definition |
|---|---|
| $\mathcal{G}$ | Graph data. |
| $\mathcal{V}$ | The node set of $\mathcal{G}$. |
| $\mathcal{E}$ | The edge set of $\mathcal{G}$. |
| $\mathbf{X}$ | The feature matrix of $\mathcal{G}$. |
| $\mathbf{A}$ | The adjacency matrix of $\mathcal{G}$. |
| $\boldsymbol{P}(\cdot)$ | The proxy model. |
| $F$ | The dimension of the node feature. |
| $K$ | The number of clients. |
| $\hat{\mathcal{G}}^k$ | The generated graph for client $k$. |
| $\hat{\mathbf{X}}$ | The pseudo feature matrix. |
| $\mathbf{H}$ | The activated similarity matrix. |
| $\hat{\mathbf{A}}$ | The pseudo adjacency matrix. |
| $v_i$ | Node $i$ in $\mathcal{V}$. |
| $h_i^l$ | The representation of $v_i$ at the $l$-th layer of GNN. |
| $\mathcal{N}(v_i)$ | The set of neighbours of node $v_i$. |
| $\Gamma$ | The transport plan in Fused Gromov-Wasserstein distance. |
| $\mathcal{G}_{\mathcal{S}}^k$ | The knowledge set for client $k$. |
| $\mathcal{G}_{\mathcal{S}}$ | The integrated knowledge set. |
| $\gamma_{ij}^l$ | The Coherence Factor at the $l$-th layer between node $v_i$ and $v_j$. |
| $\mathcal{T}_l^k$ | The Topology-Consistency Criterion of $\mathcal{G}_{\mathcal{S}}^k$ at the $l$-th layer. |
| $\boldsymbol{\theta}_{\boldsymbol{w}}^l$ | The weight matrix of the $l$-th layer. |

# B. Dataset Details

We evaluate GHOST on seven real-world graph datasets: Cora, CiteSeer, PubMed, Chameleon, Amazon-Photo, CoAuthor-CS, and Ogbn-Arxiv. For all datasets, we use a common split of 20%/40%/40% for training/validation/testing sets. The statistics for these datasets are provided in Tab. 5. The details are as follows:

- **Cora, CiteSeer, and PubMed.** These three datasets are widely used benchmarks in graph-based machine learning, particularly for node classification and link prediction tasks. Each dataset consists of a citation network where nodes represent scientific papers and edges represent citation relationships between them. The nodes are categorized into multiple classes, and each paper is represented by a feature vector derived from its content (e.g., words from titles or abstracts). These datasets are sparse, high-dimensional, and well-suited for testing models on real-world, graph-structured data. They are especially popular for evaluating graph neural networks (GNNs) and other graph-based algorithms, given their complex structure and scalability challenges.

- **Chameleon.** The Chameleon dataset represents a citation network of academic papers, where nodes correspond to papers and edges indicate citation relationships. This dataset is notable for its heterogeneous and complex community structure, containing densely connected subgraphs. These characteristics pose challenges for tasks such as community detection and graph clustering. With sparse feature vectors derived from paper content, Chameleon is often used to evaluate the performance, scalability, and robustness of graph-based models, particularly in large-scale and heterogeneous networks.

- **Amazon-Photo.** The Amazon-Photo dataset is used for node classification tasks, where nodes represent photos and edges represent co-purchase relationships between them. This dataset consists of images from the Amazon product catalog, with each photo labeled by category. Features are derived from image metadata, making the dataset suitable for evaluating graph-based models in visual domains.

- **CoAuthor-CS.** The CoAuthor-CS dataset is a citation network of computer science research papers, where nodes represent papers and edges represent co-authorship relationships. The papers are categorized into several topics, with features derived from their titles and abstracts. This dataset serves as a challenging benchmark for node classification and

community detection algorithms in academic citation networks.

- **Ogbn-Arxiv.** The Ogbn-Arxiv dataset is a large-scale citation network where nodes represent academic papers from the arXiv repository, and edges represent citation relationships. It includes node features based on paper abstracts and categorizes papers into a variety of subjects, including physics, computer science, and mathematics. Ogbn-Arxiv is frequently used to benchmark GNNs and provides a real-world challenge with diverse categories.

*Table 5.* **Statistics** of datasets used in experiments.

| Dataset | #Nodes | #Edges | #Classes | #Features |
|---|---|---|---|---|
| Cora | 2,708 | 5,278 | 7 | 1,433 |
| Citeseer | 3,327 | 4,552 | 6 | 3,703 |
| Pubmed | 19,717 | 44,324 | 3 | 500 |
| Chameleon | 2277 | 36101 | 5 | 2325 |
| Amz-Photo | 7,650 | 287,326 | 8 | 745 |
| Coauthor-CS | 18,333 | 327,576 | 15 | 6,805 |
| Obgn-Arxiv | 169,343 | 1,166,243 | 40 | 128 |

## C. Baseline Details

In this section, we make descriptions of all the baseline methods in our experiment.

- **FedAvg** [ASTAT17]. FedAvg is a widely-used baseline algorithm in Federated Learning, where local clients independently train models on their local data and then aggregate their model updates on a central server. The server computes a weighted average of the model parameters based on the clients' updates and uses this to update the global model, which is then sent back to the clients for further training. This method reduces communication overhead by transmitting only model updates instead of raw data, making it suitable for privacy-preserving distributed learning (Li et al., 2019; Mehta & Aneja, 2024). However, its performance can degrade in scenarios with heterogeneous data distributions across clients.

- **FedProx** [MLSys20]. FedProx is an enhanced version of the FedAvg algorithm, designed to address the challenges posed by data heterogeneity (Non-IID data) in federated learning. In FedProx, each client optimizes its local model with an additional regularization term that penalizes large deviations from the global model. This proximal term helps to mitigate the negative effects of local data distributions that may differ significantly from the global data distribution, thus improving the convergence of the global model (Yuan & Li, 2022; Li & Richtárik, 2024). By balancing the local training process with the global model, FedProx ensures that each client's updates are more consistent with the global model, resulting in better model performance and stability, particularly in the presence of non-IID data across clients.

- **FedNova** [NeurIPS20]. FedNova is another extension of the FedAvg algorithm that aims to improve the performance of federated learning under data heterogeneity. Unlike FedAvg, which averages model updates directly, FedNova normalizes the local updates before aggregation. This normalization ensures that the contribution of each client to the global model is proportional to the scale of its local data. By normalizing the local gradients or updates, FedNova addresses issues such as the unequal influence of clients with varying amounts of data, leading to more stable and efficient convergence. This approach is particularly useful in federated learning scenarios where clients' data distributions are highly skewed, as it ensures that the global model is better aligned with the diversity of the local data.

- **FedRCL** [CVPR24]. FedRCL is a novel approach designed to tackle data heterogeneity challenges in FL with contrastive learning (You et al., 2020; Wan et al., 2024b). It begins by analyzing the inconsistencies in gradient updates across clients during local training, which are found to be influenced by the distribution of feature representations. FedRCL introduces a relaxed contrastive learning loss that applies a divergence penalty to excessively similar sample pairs within each class, preventing collapsed representations. This strategy improves feature transferability and enhances collaborative learning across clients. As a result, FedRCL leads to significant performance improvements over existing federated learning methods, as demonstrated by extensive experimental results on standard benchmarks.

- **FedPub** [ICML23]. FedPub is designed to address the challenges of heterogeneity in subgraph-based graph learning. Unlike traditional methods that focus on learning a single global model, FedPub emphasizes the joint improvement of local GNN in a personalized manner. It utilizes functional embeddings to compute similarities between local GNNs, enabling weighted averaging for server-side aggregation. Additionally, FedPub introduces a personalized sparse mask for each client to select and update only the subgraph-relevant parameters. This approach effectively handles data privacy and heterogeneity, outperforming existing methods on various subgraph datasets.

- **FedTAD** [IJCAI24]. FedTAD is a novel approach designed to address the challenge of subgraph heterogeneity in subgraph federated learning . In subgraph-FL, the variation in node attributes and graph topology across different clients can impair the performance of the global GNN. FedTAD decouples these variations into label distribution differences and structure homophily, revealing how they affect the reliability of knowledge from local models. This variation results in misguiding model aggregation, as local models contribute differently depending on the class-wise knowledge reliability. FedTAD enhances the knowledge transfer from local models to the global model by utilizing topology-aware data-free knowledge distillation, which improves the reliability and efficiency of the aggregation process.
- **FedGTA** [VLDB24]. FedGTA is a personalized optimization strategy designed to address the challenges in FGL, particularly in handling large-scale subgraphs and improving performance. While existing FGL approaches either focus on optimization for multi-client training or complex local models, they often fall short due to slow convergence and scalability issues, especially when applied to graph data. FedGTA optimizes FGL through topology-aware local smoothing and mixed neighbor features, taking into account the graph structure for more efficient aggregation (Zhu et al., 2024a). FedGTA is among the first to bridge large-scale graph learning with federated learning, contributing significantly to the development of scalable FGL methods.
- **DENSE** [NeurIPS22]. DENSE is a novel framework designed to address the limitations of existing one-shot Federated Learning (FL) methods, which typically require additional data or model information and often assume homogeneous client models. DENSE overcomes these challenges by utilizing a two-stage process: a data generation stage and a model distillation stage. This approach enables the global model to be trained in a single communication round, without requiring extra data or model information to be transferred between clients and the server. Additionally, DENSE accommodates model heterogeneity, allowing clients to use different model architectures.
- **FedCVAE** [ICLR23]. FedCVAE is a data-free one-shot FL method designed to handle high statistical heterogeneity and improve security. It utilizes a Conditional Variational Autoencoder (CVAE) (Doersch, 2016; Kipf & Welling, 2016; Kim et al., 2021) to reframe the local learning task, enabling effective model aggregation despite differences in local data distributions. An extended version, FEDCVAE-KD, incorporates knowledge distillation (Gou et al., 2021; Bi et al., 2025b;a) to compress local decoders into a single global decoder, further enhancing performance. FedCVAE does not require additional data or auxiliary datasets, supports heterogeneous local models, and improves security by shifting the CVAE prior distribution. It significantly outperforms traditional methods, especially under conditions of high heterogeneity. In our experiment, we utilize FedCVAE-ENS as the baseline.
- **FedSD2C** [NeurIPS24]. FedSD2C is a novel one-shot FL framework designed to address the challenges of data heterogeneity and scalability in federated settings. While one-shot FL offers advantages in communication efficiency and privacy preservation, it often compromises model performance due to inconsistent local data distributions and information loss during knowledge transfer. FedSD2C tackles these issues by introducing a distiller that synthesizes informative distillates directly from local data, reducing the two-step information loss typically seen in traditional methods. Instead of aggregating inconsistent local models, FedSD2C shares these synthetic distillates, ensuring better consistency and more reliable knowledge transfer to the global model.

## D. Implementation Details

The experiments are conducted using NVIDIA GeForce RTX 3090 GPUs as the hardware platform, coupled with Intel(R) Xeon(R) CPU E5-2678 v3 @ 2.50GHz. The deep learning framework employed was Pytorch, version 2.3.1, alongside CUDA version 12.1. We adopt a two-layer GCN as the backbone, with the hidden layer size as 128. Moreover, we utilize 3 hidden linear layers and a projection head as the proxy model which concats random noise and one-hot label as the input and generates pseudo features as the output. We set draw $p_k \sim \text{Dir}(\alpha)$ from a Dirichlet distribution (Minka, 2000) and assign a fraction $p_k^c$ of class $c$ to client $k$. As for optimization of the proxy model, Adaptive Moment Estimation (Adam) (Kingma, 2014) was chosen, featuring a learning rate of $5e-3$ and a weight decay of $4e-4$. For the alignment phase of each proxy model of 10 clients, we set the local training epoch $T_L$ to 100. $\lambda_d$ and $\lambda_f$ are determined through a grid search (Liashchynskyi & Liashchynskyi, 2019) within $\{0.01, 0.05, 0.1, 0.5\}$ and $\{0.1, 0.2, 0.5, 1\}$ respectively. To make sure that $\mathcal{L}_{fgw}$ is on the same scale as other loss functions for Amz-Photo and Ogbn-Arxiv datasets, we set their $\lambda_f$ scales to $1e-4$ and $1e-7$, respectively. We set $a$ in $\mathcal{L}_{fgw}$ as 0.5 to balance the feature part and the structure part. The communication round is limited to **one**. At the server side, we set $M=3$, $T_G=5$ and adopt Adam as the optimizer for the global model with a learning rate of $1e-2$ and a weight decay of $4e-4$. As for $\lambda_r$ and $\lambda_n$, we conduct a grid search within $\{0.1, 0.5, 1\}$ respectively. The pseudo graphs generated by the proxy model of each client have the same scale as the corresponding local subgraph. Considering the huge amount of model parameters, we set the scale for $\lambda_r$ and $\lambda_n$ as $1e-11$ to balance each loss function. As shown in Sec. 5.4, all the components and modules make significant contributions to the performance

---

**Algorithm 1** The framework of GHOST

---

**Input**: Initial Proxy model of the $k$-th client $\boldsymbol{P}^k(\cdot)$, random noise $z$ , local training set $\mathcal{G}^k$ and labels $y^k$, local training epoch $T_L$, initial global model $\mathcal{M}(\cdot)$, number of layers $L$, global training epoch $T_G$, hyperparameters $\lambda_d, \lambda_f, \lambda_r$ and $\lambda_n$.

 1: **for** each client $k$ **do**
 2:     **for** local epoch $t \leftarrow 1, \ldots, T_L$ **do**
 3:         Input $z$ and $y^k$ to $\boldsymbol{P}^k(\cdot)$;
 4:         Generate pseudo graph $\hat{\mathcal{G}}^k$ through Equation (2) and Equation (3);
         */* Dual-Level Alignment for the proxy model */*
 5:         Compute $\mathcal{L}_{disp}, \mathcal{L}_{div}, \mathcal{L}_{fgw}$ between $\hat{\mathcal{G}}^k$ and $\mathcal{G}^k$;
 6:         $\mathcal{L}_{dual} = \mathcal{L}_{disp} + \lambda_d \mathcal{L}_{div} + \lambda_f \mathcal{L}_{fgw}$;
 7:         Update local proxy model via $\nabla \mathcal{L}_{dual}$.
 8:     **end for**
 9: **end for**
10: Each client uploads parameters of its proxy model to the server.
    */* Knowledge Integration */*
11: **for** each proxy model $\boldsymbol{P}^k(\cdot)$ **do**
12:     Input $z$ and uniform pseudo labels $\hat{y}^k$;
13:     Generate knowledge set $\mathcal{G}_S^k$ of client $k$ by $\boldsymbol{P}^k(\cdot)$.
14: **end for**
15: Generate the integrated knowledge set $\mathcal{G}_S$.
    */* Global Training */*
16: **for** pseudo graphs $\hat{\mathcal{G}}_1^k, \ldots, \hat{\mathcal{G}}_M^k$ in $\mathcal{G}_S^k \in \mathcal{G}_S$ **do**
17:     **for** global epoch $t \leftarrow 1, \ldots, T_G$ **do**
18:         Input $\hat{\mathcal{G}}_1^k, \ldots, \hat{\mathcal{G}}_M^k$ to global model $\mathcal{M}(\cdot)$;
19:         Compute $\mathcal{L}_{task}^k$.
        */* Topology Knowledge Retention */*
20:         **for** $l$-th hidden layer **do**
21:             Compute $\mathcal{T}_l^k$ through Equation (9).
22:         **end for**
23:         Compute $\mathcal{L}_r^k$ through Equation (10);
24:         $\mathcal{L}_{global}^k = \mathcal{L}_{task}^k + \lambda_r \mathcal{L}_r^k + \lambda_n ||\mathcal{T}^k||_2$;
25:         Update global model via $\nabla \mathcal{L}_{global}^k$.
26:     **end for**
27: **end for**

---

enhancement. Furthermore, we provide the detailed description of our framework in Algorithm 1.

## E. Ablation Study on Different Numbers of Clients.

In this section, we vary the number of clients in $\{5, 10, 20\}$ and conduct the node classification task on CiteSeer, Chameleon, Amazon-Photo and Coauthor-CS datasets. Experimental Results are shown in Tab. 6. From the table, we can observe that our GHOST outperforms most baselines with different numbers of clients, demonstrating the stability of GHOST across various data distributions and subgraph scales.

## F. Sensitivity

To address **Q4**, we conduct analysis on hyperparameters of GHOST. Specifically, we compare the node classification performance under different values of $\lambda_d$ and $\lambda_f$. We take Cora, Coauthor-CS as example datasets. For both datasets, we vary $\lambda_d$ and $\lambda_f$ in range $[0.01, 0.09]$ and $[0.05, 0.25]$ with 0.02 and 0.05 as step size respectively. Experimental results are shown in Figure 4 and Figure 6. From the bar charts, we can observe that the performance is not influenced much at different values. Moreover, all studies of $\lambda_d$ and $\lambda_f$ outperform the baseline, proving the robustness and stability of GHOST.

*Table 6.* **Comparison with the state-of-the-art methods with different numbers of clients.** We report node classification accuracies (%) for downstream task performance. Green arrows ↑ denote advancements in accuracy metrics than FedAvg while red arrows ↓ indicate regressions. OOM means out-of-memory error. The best and second results are highlighted with **bold** and underline, respectively.

| Datasets (→) | CiteSeer | | | Chameleon | | |
|---|---|---|---|---|---|---|
| Methods (↓) | 5 Clients | 10 Clients | 20 Clients | 5 Clients | 10 Clients | 20 Clients |
| FedAvg [ASTAT17] | 35.43 | 32.88 | 38.13 | 21.31 | 19.89 | 20.02 |
| Traditional FL | | | | | | |
| FedProx [MLSys20] | $40.82_{\uparrow 5.39}$ | $35.73_{\uparrow 2.85}$ | $39.39_{\uparrow 1.26}$ | $20.77_{\downarrow 0.54}$ | $19.78_{\downarrow 0.11}$ | $19.81_{\downarrow 0.21}$ |
| FedNova [NeurIPS20] | $18.43_{\downarrow 17.00}$ | $18.58_{\downarrow 14.30}$ | $16.17_{\downarrow 21.96}$ | $20.33_{\downarrow 1.01}$ | $21.30_{\uparrow 1.41}$ | $17.75_{\downarrow 2.27}$ |
| FedRCL [CVPR24] | $15.88_{\downarrow 19.55}$ | $12.73_{\downarrow 20.15}$ | $7.05_{\downarrow 31.08}$ | $26.89_{\uparrow 5.58}$ | $18.48_{\downarrow 1.41}$ | $20.13_{\uparrow 0.11}$ |
| Traditional FGL | | | | | | |
| FedPub [ICML23] | $20.07_{\downarrow 15.36}$ | $34.91_{\uparrow 2.03}$ | $32.27_{\downarrow 5.86}$ | $20.33_{\downarrow 0.98}$ | $17.61_{\downarrow 2.28}$ | $23.16_{\uparrow 3.14}$ |
| FedTAD [IJCAI24] | $19.25_{\downarrow 16.18}$ | $33.86_{\uparrow 0.98}$ | $28.34_{\downarrow 9.79}$ | $28.01_{\uparrow 6.70}$ | $20.11_{\uparrow 0.22}$ | $23.05_{\uparrow 3.03}$ |
| FedGTA [VLDB24] | $16.93_{\downarrow 18.50}$ | $17.75_{\downarrow 15.13}$ | $15.73_{\downarrow 22.40}$ | $20.32_{\downarrow 0.99}$ | $21.20_{\uparrow 1.31}$ | $19.81_{\downarrow 0.21}$ |
| One-shot FL | | | | | | |
| DENSE [NeurIPS22] | $7.64_{\downarrow 27.79}$ | $7.87_{\downarrow 25.01}$ | $7.06_{\downarrow 31.07}$ | $20.00_{\downarrow 1.31}$ | $19.90_{\uparrow 0.01}$ | $20.13_{\uparrow 0.11}$ |
| FedCVAE [ICLR23] | $42.64_{\uparrow 7.21}$ | $34.76_{\uparrow 1.88}$ | $18.10_{\downarrow 20.03}$ | $19.89_{\downarrow 1.42}$ | $21.74_{\uparrow 1.85}$ | $16.99_{\downarrow 3.03}$ |
| FedSD2C [NeruIPS24] | $20.97_{\downarrow 14.46}$ | $29.96_{\downarrow 2.92}$ | $23.66_{\downarrow 14.47}$ | $25.14_{\uparrow 3.83}$ | $22.72_{\uparrow 2.83}$ | **$24.35_{\uparrow 4.33}$** |
| GHOST | **$42.92_{\uparrow 7.49}$** | **$37.75_{\uparrow 4.87}$** | **$40.80_{\uparrow 2.67}$** | **$28.42_{\uparrow 7.11}$** | **$25.22_{\uparrow 5.33}$** | $23.59_{\uparrow 3.57}$ |
| Datasets (→) | Amazon-Photo | | | Coauthor-CS | | |
| Methods (↓) | 5 Clients | 10 Clients | 20 Clients | 5 Clients | 10 Clients | 20 Clients |
| FedAvg [ASTAT17] | 49.93 | 23.12 | 21.79 | 12.84 | 22.50 | 33.51 |
| Traditional FL | | | | | | |
| FedProx [MLSys20] | $46.54_{\downarrow 3.39}$ | $24.16_{\uparrow 1.04}$ | $23.58_{\uparrow 1.79}$ | $16.25_{\uparrow 3.41}$ | $21.44_{\downarrow 1.06}$ | $29.33_{\downarrow 4.18}$ |
| FedNova [NeurIPS20] | $9.69_{\downarrow 40.24}$ | $6.15_{\downarrow 16.97}$ | $8.57_{\downarrow 13.22}$ | $22.45_{\uparrow 1.14}$ | $18.83_{\downarrow 3.67}$ | $16.37_{\downarrow 17.14}$ |
| FedRCL [CVPR24] | $21.80_{\downarrow 28.13}$ | $4.92_{\downarrow 18.20}$ | $10.69_{\downarrow 11.10}$ | $8.33_{\downarrow 4.51}$ | $14.75_{\downarrow 7.75}$ | $7.02_{\downarrow 26.49}$ |
| Traditional FGL | | | | | | |
| FedPub [ICML23] | $42.00_{\downarrow 7.93}$ | $21.91_{\downarrow 1.21}$ | $21.76_{\downarrow 0.03}$ | $5.14_{\downarrow 7.70}$ | $26.75_{\uparrow 4.25}$ | $22.55_{\downarrow 10.96}$ |
| FedTAD [IJCAI24] | $25.29_{\downarrow 24.64}$ | $22.01_{\downarrow 1.11}$ | $21.57_{\downarrow 0.22}$ | $11.77_{\downarrow 1.07}$ | $14.09_{\downarrow 8.41}$ | $36.42_{\uparrow 2.91}$ |
| FedGTA [VLDB24] | $5.39_{\downarrow 44.54}$ | $4.10_{\downarrow 19.02}$ | $6.17_{\downarrow 15.62}$ | $10.16_{\downarrow 2.68}$ | $10.80_{\downarrow 11.70}$ | $9.43_{\downarrow 24.08}$ |
| One-shot FL | | | | | | |
| DENSE [NeurIPS22] | $4.96_{\downarrow 44.97}$ | $4.93_{\downarrow 18.19}$ | $4.90_{\downarrow 16.89}$ | $3.75_{\downarrow 9.09}$ | $3.96_{\downarrow 15.93}$ | $3.71_{\downarrow 29.80}$ |
| FedCVAE [ICLR23] | $52.39_{\uparrow 2.54}$ | $31.62_{\uparrow 8.50}$ | $24.87_{\uparrow 3.08}$ | $2.56_{\downarrow 10.28}$ | $14.60_{\downarrow 7.90}$ | $10.98_{\downarrow 22.53}$ |
| FedSD2C [NeruIPS24] | $11.75_{\downarrow 38.18}$ | $8.73_{\downarrow 14.39}$ | $23.38_{\uparrow 1.59}$ | $3.95_{\downarrow 8.89}$ | $3.88_{\downarrow 18.62}$ | $9.94_{\downarrow 23.57}$ |
| GHOST | **$53.07_{\uparrow 3.14}$** | **$37.22_{\uparrow 14.10}$** | **$37.50_{\uparrow 15.71}$** | **$24.99_{\uparrow 12.15}$** | **$29.91_{\uparrow 7.41}$** | **$40.84_{\uparrow 7.33}$** |

## G. Privacy Security.

Privacy security plays a crucial role in FGL systems (Zeng et al., 2022; Liao et al., 2025). In this section, we conduct experiments on two possible approaches for the eavesdropping attacker on datasets Cora and CiteSeer, assuming the worst case where the attacker intercepts all the parameters (Li et al., 2021; Mothukuri et al., 2021). As illustrated in Sec. 4.4, without knowing the specific shift $\epsilon$, the attacker can only randomly select a center $\hat{\chi}$ away from $(\chi + \epsilon)$, or make a guess based on a uniform distribution with wide bounds. Firstly, we set the original noise distribution as standard normal distribution ($\chi = 0$) with shift $\epsilon$ varies in $\{10, 20, 30\}$ while the attacker still takes $\hat{\chi} = 0$ as an attempt. Experimental results are shown in Figure 5(a). Then, we set $(\chi + \epsilon)$ as 0 and set several guesses of uniform distribution that attempt to overlap the original distribution range: $\{\mathcal{U}(-50, 50), \mathcal{U}(-100, 100), \mathcal{U}(-150, 150)\}$. Experimental results are shown in Figure 5(b), in which we utilize bounds = 50 to denote the guess $\mathcal{U}(-50, 50)$. From the bar charts, we observe that regardless of whether the attacker guesses the center or attempts to overlap the distribution over a wide range, the pseudo-graphs generated lead to a performance drop of 50%-80%. This significant drop demonstrates that the generated pseudo-graphs differ drastically from the real data held by clients, thereby preventing any leakage of private information. Consequently, the privacy security of our GHOST framework is confirmed.

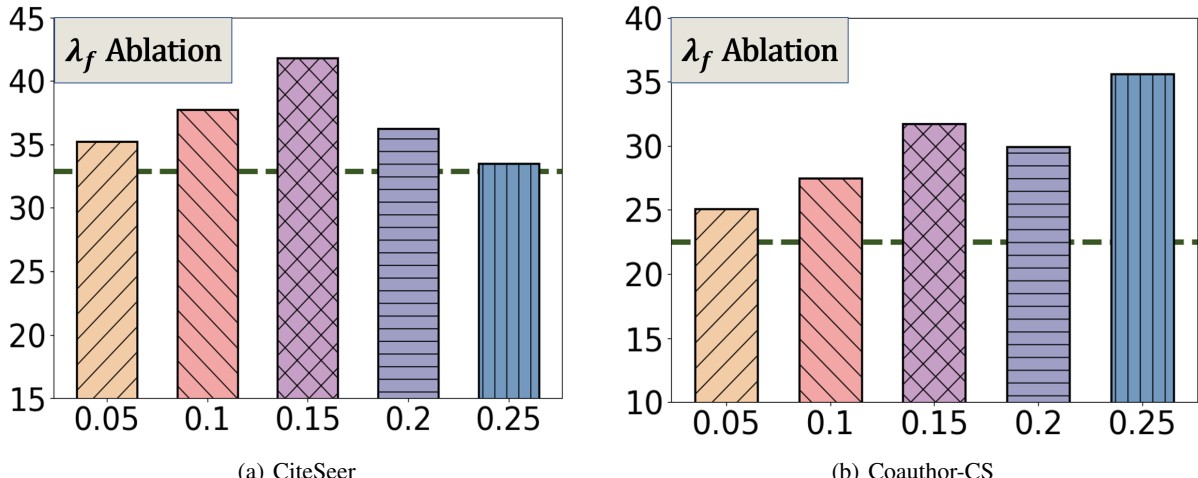

*Figure 6.* **Hyperparameter Analysis** on $\lambda_f$ on datasets CiteSeer and Coauthor-CS. The green dash denotes the baseline performance. See details in Appendix F.

## H. Discussion on Limitations.

Although GHOST has demonstrated significant success in efficiently capturing diverse dual-level knowledge and mitigating catastrophic forgetting in the one-shot setting, it still faces some limitations as a proxy-based approach (Tan et al., 2024a; Bi et al., 2025c). Specifically, noise in local data (Kang et al., 2019) can interfere with the ability of the proxy model to mine and learn local-specific knowledge, potentially hindering the training of the global model. Enhancing the robustness of the proxy model against such interference (Lu et al., 2024; Li et al., 2024a) presents an interesting direction for our future research.

