# OpenReview forum: "GHOST: Generalizable One-Shot Federated Graph Learning with Proxy-Based Topology Knowledge Retention"
_ICML.cc/2025/Conference — ICML 2025 poster_

### Official Review · Reviewer_TQDx · 2025-03-03

**Overall Recommendation:** 4

**Summary:**

This paper proposes GHOST, a novel generalized one-shot FGL framework designed to address challenges related to generalizable capability and catastrophic forgetting. The method involves two main components: Dual-Level Aligned Proxy Model and Topology-Conscious Knowledge Retention. In the first component, each client constructs a proxy model that captures local feature and topological information. In the second component, the server aggregates the proxy models from all clients to form a global model. A Topology-Consistency Criterion is applied during global training to stabilize key parameters and retain important topological information, thus mitigating catastrophic forgetting. The framework also incorporates a privacy-preserving mechanism to secure client data. Experimental results demonstrate that GHOST outperforms existing methods and shows resilience to data heterogeneity.

**Claims And Evidence:**

This paper proposes a persuasive explanation of its claims. One-shot Federated Learning has emerged as an effective approach in environments with limited resources, and there indeed is a gap between One-shot FL and traditional FGL, especially in the generalized setting, as claimed in the paper. The figure of the problem illustration clearly presents the phenomenon.

**Essential References Not Discussed:**

There are no related works that are essential to understanding key contributions of the paper, but are not currently cited in the paper.

**Experimental Designs Or Analyses:**

I have checked the soundness of the experimental designs and analyses in this paper. In the experimental part of this paper, the authors conducted the node classification task on seven real-world datasets. The privacy security part is meaningful and convincing to me. Moreover, the authors make detailed analyses on the experimental results in Sec 5.2.

**Methods And Evaluation Criteria:**

The proposed method and evaluation criteria make sense for the problem. The Dual-Level Aligned Proxy Model and Topology-Conscious Knowledge Retention are both reasonable in the one-shot setting. The node classification task is widely used as a downstream task in FGL methods and is practically applicable.

**Other Comments Or Suggestions:**

I have no other comments or suggestions.

**Other Strengths And Weaknesses:**

- Strengths:
1. The proposed method of generalized one-shot FGL appears interesting and innovative.
2. The paper provides a clear and detailed explanation of the methodology, making the method easy to follow.
3. This paper considers the structure attributes in graph data, introducing Topology-Consistency Criterion to mitigate catastrophic forgetting, which is an important addition to the field and is well-supported by both theory and experiments.
4. The inclusion of a privacy analysis is a strong point, particularly in FL where data privacy is crucial. The Center-Shifting method provides a practical solution to safeguard client data.

- Weaknesses:
1. This paper does not provide implementation details for applying one-shot FL methods to graph data. Specifically, the one-shot baselines used in the experiments, such as FedCVAE and FedSD2C, rely on generative techniques designed for image data, which differs significantly from graph data. The paper does not address how graph data are generated in these methods.
2. How does the method perform under higher levels of data heterogeneity? For example, what happens when the Dirichlet distribution parameter α is set to lower values (e.g., α=0.01)?

**Questions For Authors:**

See Weaknesses above.

**Relation To Broader Scientific Literature:**

The proposed method pushes the boundaries of FGL by offering an innovative generalized one-shot approach, filling gaps left by previous studies and paving the way for more practical applications in FGL. Moreover, the Fused Gromov-Wasserstein distance [1] in the alignment part is novel and effective.

[1] Vayer, T., Chapel, L., Flamary, R., Tavenard, R., & Courty, N. (2020). Fused Gromov-Wasserstein distance for structured objects. Algorithms, 13(9), 212.

**Theoretical Claims:**

I have checked the correctness of proofs for theoretical claims in this paper. The dual-level alignment loss can indeed capture fine-grained knowledge, thus achieving a good alignment.

---

> ### Author Rebuttal · Authors · 2025-03-28
>
> ***Dear Reviewer TQDx:***
>
> We greatly appreciate your positive feedback on our work, as well as the thoughtful concerns and questions you raised. We have carefully considered each of your comments and provided detailed responses.
>
> **Weakness:**
>
> **W1: Implementation details for applying one-shot FL methods to graph data.**
>
> We sincerely appreciate the reviewer’s suggestion to clarify the implementation details of applying one-shot FL methods to graph data. In our experiments, we follow the overall process of the compared one-shot FL methods, utilizing the encoders and decoders from the corresponding papers to generate graph data features. Additionally, we adopt the k-nearest-neighbor (KNN) method to construct the topological structure for the generated graph data.
>
> Specifically, in FedCVAE, we adhere to the Conditional Variational Autoencoder framework as used in the original paper. We input the training data and labels into the encoder to obtain representations in the latent space, which are then fed into the decoder to generate features. The KNN algorithm is subsequently applied to construct the topology. Similarly, in FedSD2C, we follow the original paper’s Autoencoder framework (including both the encoder and decoder) for generating graph data features and use the KNN algorithm for topology construction.  By doing so, we retain the key processes and steps of the original baselines while reasonably adapting them to graph data.
>
>
> **W2: Performance under higher levels of data heterogeneity (e.g. $\alpha = 0.01$)**
>
> To demonstrate the performance of our method under higher data heterogeneity, we conduct experiments on the CiteSeer, and Coauthor-CS datasets with the Dirichlet distribution parameter $\alpha$ set to 0.01. We select five baselines for comparison: FedAVG, FedNova, FedPub, FedTAD and FedCVAE. The experimental results are shown in Table 1.
>
> *Table 1: Performance of GHOST under higher data heterogeneity ($\alpha = 0.01$) on CiteSeer and Coauthor-CS datasets.*
>
> | **Methods**   | **CiteSeer** | **Coauthor-CS** |
> |---------------|--------------|-----------------|
> | FedAVG    | 23.15        | 11.09           |
> | FedNova   | 19.12        | 16.94           |
> | FedPub   | 20.71        | 11.28           |
> | FedTAD    | 21.21        | 11.19           |
> | FedCVAE   | 23.53        | 8.97            |
> | **GHOST**     | **25.32**    | **17.61**       |
>
>
> From the table, we can observe that our method maintains the best performance even under higher data heterogeneity. For some datasets, when $\alpha$ is smaller, certain clients have very few nodes, which does not align with real-world scenarios. Therefore, we set $\alpha = 0.05$ in the experiments, as it ensures high data heterogeneity while avoiding overly extreme scenarios that do not reflect real-world conditions.

---

> > ### Comment · Reviewer_TQDx · 2025-04-03
> >
> > Thank you for the response. The additional experiments under higher data heterogeneity resolve my questions. I will maintain my positive score.

---

> > > ### Author Response · Authors · 2025-04-03
> > >
> > > ***Dear Reviewer TQDx:***
> > >
> > > Thank you for taking the time to revisit our work. We are pleased that our clarifications and additional results have addressed your questions. Your positive evaluation means a lot to us.
> > >
> > > Best Regards,
> > >
> > > Authors

---

### Official Review · Reviewer_9vCa · 2025-03-04

**Overall Recommendation:** 4

**Summary:**

This work focuses on issues of communication overload, generalizability and catastrophic forgetting and proposes a one-shot approach where each client constructs a proxy model After the alignment, these models are then uploaded to the server to train a global model in an ensemble manner. This work also adopts topology knowledge retention to mitigate catastrophic forgetting. Privacy protection is also a key component in this work, ensuring the security of client data. Experimental results indicate that GHOST outperforms existing methods, showing robust performance in the face of data heterogeneity.

**Claims And Evidence:**

This work identifies the challenges of poor generalizable ability and catastrophic forgetting and proposes clear evidence in the introduction part.

**Essential References Not Discussed:**

None.

**Experimental Designs Or Analyses:**

The experimental designs and analyses in this work are soundness to me. Performance on  seven datasets with various scales shows the superiority of this work. The ablation study and sensitivity study are sufficient, making strong proof for the contribution of each module. In Appendix D, the authors conduct the experiment with different client scales, showing the resilience of this work. However, I have one issue about the results, which will be proposed in details in **Questions For Authors:** part.

**Methods And Evaluation Criteria:**

The framework figure (Figure 2) makes clear and vivid elaboration of the overall method. All the modules in the work are closely related to the identified problems. The benchmark datasets are highly relevant with the problem as well.

**Other Comments Or Suggestions:**

None.

**Other Strengths And Weaknesses:**

Strengths:

1. Novel and well-motivated. This work points out the issue of communication overload and innovatively bridges the gap between FGL and one-shot setting. With efficient aligning and knowledge retention, this work effectively addresses the problems left by previous studies.

2. Easy to follow. The paper is well-written with a detailed and exquisite framework visualization. Moreover, the precise descriptions of modules in the methodology makes the overall work clear and easy to follow.

3. Topology-aware. The topology knowledge retention module is meaningful. Graph data significantly differ from image data due to their structure attribute in non-Euclidean space. This work highlights the importance of graph structure and strive to consolidate the topology knowledge during training, which makes sense.

Weaknesses:

1. Lack of further explanation of the pseudo labels: This work assumes that the pseudo labels have a uniform distribution across all classes. This assumption could be unrealistic in some real-world datasets, where the class distributions might be skewed.

2. Lack of more sensitivity study. The work lacks analysis of model performance with different local training epochs of each proxy model.

3. Lack of training details: The work lacks details of the local training epoch for other traditional FL/FGL or one-shot FL baselines in the experimental section.

**Questions For Authors:**

Model performance with varying client numbers: I noticed the experiment in Appendix D, which evaluates the model's performance with different client scales. Why does the model perform better with more clients on some datasets (e.g., Coauthor-CS)? With 5 clients, each client should have more training data compared to when there are 10 clients, so why does the model perform worse under the former condition (24.99 for 5 clients vs. 29.91 for 10 clients)?

**Relation To Broader Scientific Literature:**

GHOST represents a novel step forward in the field of FGL by presenting a unique one-shot approach. It overcomes some of the key limitations found in previous studies, creating new avenues for future research.

**Theoretical Claims:**

Both the proxy model alignment part and topological knowledge retention part are supported by rigorous analyses. The identification of parameters crucial for capturing topological knowledge is interesting, where the topological attributes at the hidden layers are fully explored and captured. I believe topological information is of great significance and is worth highlighting in graph learning.

---

> ### Author Rebuttal · Authors · 2025-03-28
>
> ***Dear Reviewer 9vCa:***
>
> We sincerely appreciate the time and effort you have dedicated to reviewing our paper. We hope that the detailed responses provided below will effectively address your concerns and offer the necessary clarifications.
>
> **Weakness：**
>
> **W1: Lack of further explanation of the pseudo labels.**
>
> We appreciate the reviewer’s valuable comment regarding the assumption of a uniform distribution for the pseudo labels. While we acknowledge that real-world datasets may exhibit skewed class distributions, we believe that the uniform distribution assumption provides several benefits in our context.
>
> Firstly, adopting a uniform distribution avoids the need for clients to share class label distributions, which significantly mitigates the risk of data privacy leakage. This aligns with the principles of privacy preservation in Federated Graph Learning, ensuring that sensitive label information is not exposed during the training process.
>
> Secondly, the use of a uniform distribution ensures that data from all classes are taken into account, including minor or underrepresented classes. This prevents the model from neglecting the features and information from minority class data, thus promoting a more balanced learning process.
>
> We believe that this design choice strikes a good balance between privacy preservation and model performance, and the effectiveness of this approach is supported by the results presented in the paper.
>
> **W2: Lack of more sensitivity study on different local training epoches.**
>
> To demonstrate the robustness of our method under different local training epochs, we select Cora and CiteSeer datasets, and design the range of local training epochs as [90, 110] with a step size of 5. Experimental results are shown in Table 1 and Table 2.
>
> *Table 1: Different local training epoches on the Cora dataset.*
> | Epoches |  90 | 95 | 100 | 105 | 110 | Avg
> |----------|--------|--------|--------|---------|-----------|-------|
> |FedAVG | 30.52 | 30.55| 30.61 | 30.62 | 30.61 | 30.58
> |FedTAD | 30.09 | 30.21 | 30.43 | 31.15 | 30.43 | 30.46
> |FedCVAE | 26.86 | 21.72 | 30.89 | 29.61 | 32.36 | 28.29
> |**GHOST** | **46.93** | **47.39** | **50.41** | **47.02** | **47.57** | **47.86**
>
> *Table 2: Different local training epoches on the CiteSeer dataset.*
> | Epoches |  90 | 95 | 100 | 105 | 110 | Avg
> |----------|--------|--------|--------|---------|-----------|-------|
> |FedAVG |  32.73 | 33.03 | 32.88 | 32.36 | 32.66 | 32.73
> |FedTAD | 20.52 | 23.30 | 33.86 | 23.75 | 25.62 | 24.72
> |FedCVAE | 27.19 | 33.56 | 34.76 | 32.13 | 24.19 | 30.37
> |**GHOST** | **40.90** | **40.52** | **37.75** | **40.62** | **37.68** | **39.49**
>
> From the tables, we can observe that our method maintains strong performance across different epochs, demonstrating its stability under varying epoches.
>
> **W3: Lack of details of training epoches in other baselines**
>
> For all the baselines compared in the experimental section, we uniformly set the local training epoch to 100 to ensure sufficient convergence of the local models. In the one-shot setting, too few local training epochs would prevent the model from adequately learning the local data, making the comparison unfair. Conversely, too many epochs could lead to overfitting to local data, resulting in degraded generalization performance.
>
> **Questions**
>
> **Q1: Model performance with varying client numbers**
>
> **A1:** We appreciate the insightful question regarding model performance under varying client numbers. The observed performance improvement with a higher number of clients, particularly in datasets with high data heterogeneity (e.g., Coauthor-CS), can be attributed to the impact of distributional bias among clients.
>
> When the number of clients is small, each client tends to hold a larger amount of data. However, in highly heterogeneous settings, this also means that the data distributions across clients can be significantly different, leading to greater heterogeneity in both data volume and feature distribution. Such imbalances can make it more challenging for the global model to effectively capture a generalizable representation, resulting in degraded performance. Conversely, increasing the number of clients leads to a finer-grained partitioning of the data, which can help mitigate extreme distribution shifts among clients. This, in turn, allows the global model to better learn shared patterns across clients, improving overall performance.
>
> Additionally, different datasets have their own unique characteristics in terms of data features and structure. As a result, the effect of varying client numbers may manifest differently across datasets, leading to dataset-specific trends in performance. We hope this clarifies the phenomenon and appreciate the opportunity to further discuss our findings.

---

> > ### Comment · Reviewer_9vCa · 2025-04-03
> >
> > I have read the authors' responses.  The explanations resolve all my previous confusion, and I've checked other reviewers' feedback as well. I will raise my score and vote for acceptance.

---

> > > ### Author Response · Authors · 2025-04-03
> > >
> > > ***Dear Reviewer 9vCa:***
> > > Thank you for your thoughtful feedback and for taking the time to review our responses. We truly appreciate your constructive insights and your willingness to engage with our clarifications. Your support and recognition of our work mean a lot to us.
> > >
> > > Best Regards,
> > >
> > > Authors

---

### Official Review · Reviewer_AgVq · 2025-03-12

**Overall Recommendation:** 4

**Summary:**

This study tackles challenges such as communication overhead, limited generalization ability, and catastrophic forgetting. It introduces a one-shot strategy where clients independently build proxy models, which are later aligned in both feature-structural level and aggregated on the server to form a global model. To address catastrophic forgetting, the approach incorporates topology-aware knowledge retention. Additionally, this study conducts adequate experiments and addresses the privacy security.

**Claims And Evidence:**

Claims made in this study are supported by clear and convincing evidence. This study precisely identifies the limitations of existing FGL methods and addresses these issues with effective and convincing approach (i.e. the “Dual-Level Alignment” module).

**Essential References Not Discussed:**

Essential related works are comprehensively cited in this study.

**Experimental Designs Or Analyses:**

I have checked the experimental designs and analyses. The authors have conducted the node classification task on graphs of various scales and make comprehensive comparisons with traditional FL/ FGL and one-shot FL methods. Moreover, the ablation study and the sensitivity study both strengthen the validity of this study.  However, I think it will be better with more FGL/ one-shot FL methods compared in the experimental part.

**Methods And Evaluation Criteria:**

The proposed methods and evaluation criteria actually make sense for the identified problems. Two key modules (the “Dual-Level Alignment” module and the “Topology Knowledge Retention” module) are both highly related to those problems.

**Other Comments Or Suggestions:**

The authors should conduct comparison experiments with more traditional FGL and one-shot FL methods.

**Other Strengths And Weaknesses:**

Strength:
S1: It is novel and meaningful to propose a one-shot FGL approach. This study precisely identifies the challenges of existing FGL methods and tackles them effectively. The overall approach is well-motivated, and the motivation is also explained with clarity.
S2: The figures of problem and framework illustration are clear and detailed, and the equations and explanations are reasonable. I believe this is a comprehensive  framework for the graph data.
S3: The proposed approach throughout the study flows seamlessly. Both two key modules and their components are innovative and coherent. The Dual-Level Aligned Proxy Model effectively captures the feature-structural knowledge and the Topology Knowledge Retention module then integrates the diverse information against the data heterogeneity while mitigating catastrophic forgetting.

Weaknesses:
W1: This study uses many notations. To improve clarity, the authors should provide a summarize table of the commonly used notations and their definitions.
W2: The authors compare the proposed approach with several traditional FL/FGL and one-shot FGL methods , but additional comparisons with more FGL and one-shot FL methods could strengthen this study.
W3: Equation 4 is designed to align the label distributions of the true and pseudo data, and Equation 5 is designed to align the feature distributions of the true and pseudo data. Essentially they serve the same purpose, so the necessity and motivation for using them simultaneously needs to be made clear.
W4: As shown in Equation 6, the optimization objective of fused GW-OT is the transport matrix T. How does Equation 6 serve a loss and yield the gradients on the pseudo feature matrix?
W5: Equation 9 is very confusing. Actually, I can’t clearly understand the meaning of it. It’s suggested to add more explanations about Equation 9.

**Questions For Authors:**

Q1: In Section 4.3, the authors propose the Topology-Conscious Knowledge Retention module and compute the Topology-Consistency Criterion to identify crucial parameters for capturing structural knowledge. Can you provide a more detailed explanation of how the Topology-Consistency Criterion reflects the importance of each parameter in learning topological information?

**Relation To Broader Scientific Literature:**

This study proposes an novel one-shot FGL approach, effectively tackling the identified issues such as limited generalization ability and catastrophic forgetting. The study is meaningful and raises the concern of the communication overhead problem in FGL systems. I believe one-shot FGL is a promising solution.

**Theoretical Claims:**

I have checked all the theoretical claims.

---

> ### Author Rebuttal · Authors · 2025-03-28
>
> ***Dear Reviewer AgVq:***
>
> Thank you for your thoughtful review and for highlighting important concerns. Below, we provide detailed responses to clarify our proposed approach.
>
> **Weaknesses & Questions：**
>
> **W1:  Notation clarity and summarization.**
>
> We sincerely appreciate the reviewer’s valuable suggestion regarding notation clarity. To enhance readability, we will incorporate a summary table of commonly used notations and their definitions in a future version of our work. Thank you for your insightful feedback.
>
> **W2 & Suggestions: Additional comparisons with more FGL and one-shot FL methods.**
>
> Thank you for your valuable suggestion. To further enhance our study, we have conducted experiments incorporating **FGSSL** (FGL) and **FENS** (one-shot FL). The results on six benchmark datasets are presented in Table 1.
>
> *Table 1: Performance comparison including additional FGL and one-shot FL methods on six benchmark datasets.*
>
> | Method      | Cora | CiteSeer | Pubmed | Amz-Photo  | Coauthor-CS  | Ogbn-Arxiv |
> |--------------|--------|--------|--------|--------|-------|-----------|
> | FGSSL    | 30.25 | 21.95 | 39.68 |  13.06 | 22.44 | 9.24
> | FENS  |  31.43 | 20.97 | 49.07 | 25.30 | 22.54 | 13.09
> | **GHOST** | **50.41** | **37.75** | **58.87** | **37.22** | **29.91** | **17.24**
>
> From the results, we observe that **GHOST** continues to achieve the best performance, further demonstrating its robustness and effectiveness. We greatly appreciate your insightful suggestion, as it has helped reinforce the comprehensiveness of our study.
>
> **W3: Necessity and motivation for using both Divergence Loss (Eq. 4) and Dispersion Loss (Eq. 5).**
>
> Thank you for raising this point. Although both losses act on the feature representations, they capture complementary aspects. The Divergence Loss minimizes the statistical discrepancy between the pseudo and real features, ensuring that overall properties (spread, variance, density) match. In contrast, the Dispersion Loss  preserves the angular (directional) relationships among features, maintaining the geometric configuration. This helps maintain the intrinsic relational patterns among the features regardless of their magnitudes. Together, they provide a comprehensive alignment, as confirmed by our ablation study (Sec. 5.4).
>
> **W4: Clarification on Equation 6 as a loss function and its gradient flow.**
>
> Thank you for this important question. Although Equation 6 formulates the FGW loss in terms of the transport plan $\mathbf{\Gamma}$, the loss function is inherently a function of both the graph structures and the feature matrices of the real and pseudo graphs.
>
> In practice, we use a differentiable optimal transport solver (implemented with a conditional gradient method with optional Armijo line-search) to obtain $\mathbf{\Gamma}$. Consequently, even though $\mathbf{\Gamma}$ is the immediate optimization variable in Equation 6, the overall FGW loss is differentiable with respect to $\hat{\mathbf{X}}$ (and $\hat{\mathbf{A}}$), allowing gradients to flow back to these pseudo graph parameters.
>
> Thus, by minimizing the FGW loss, the model not only optimizes the transport plan for aligning the local and pseudo graphs but also adjusts the pseudo feature matrix to reduce the discrepancy in both feature and structural domains. This gradient propagation through the differentiable OT solver ensures that the pseudo feature matrix is updated in a manner that preserves the intrinsic topological relationships while matching the local statistical properties.
>
> **W5 & Q1: Explanation of Equation 9 (Topology-Consistency Criterion).**
>
> Thank you for highlighting the need for clarification. The Topology-Consistency Criterion quantifies how much each global GNN parameter contributes to preserving learned topological structures. We first compute a Coherence Factor $\gamma_{ij}^l$ for connected nodes, which quantifies the similarity of their hidden representations after a transformation by the corresponding layer’s weight matrix $\boldsymbol{\theta}_{\boldsymbol{w}}^l$. Aggregating these values across nodes yields a global topological consistency measure for layer $l$. In Equation 9, the norm of the derivative for each parameter reflects its **“sensitivity”: a higher norm indicates that even a small change in that parameter would cause a significant alteration in the coherence.** In other words, parameters with high sensitivity are deemed more important for capturing the intrinsic topological structure across the diverse graphs in our integrated knowledge set.
>
> By computing and then consolidating these sensitivity measures across layers, we obtain a comprehensive criterion $\mathcal{T}^k$ that identifies which parameters are crucial for preserving structural knowledge. This process helps in selectively retaining important parameters during the global training phase, mitigating the risk of catastrophic forgetting.

---

### Decision · Program_Chairs · 2025-05-01

**Decision:**

Accept (poster)

**Comment:**

This paper is sufficient in its novelty of methodology and experimental design. The clear presentation also helps reviewers be able to fastly grasp this paper.